# Resistance of Leukemia Cells to 5-Azacytidine: Different Responses to the Same Induction Protocol

**DOI:** 10.3390/cancers15113063

**Published:** 2023-06-05

**Authors:** Kristína Šimoničová, Lubos Janotka, Helena Kavcova, Zdena Sulova, Lucia Messingerova, Albert Breier

**Affiliations:** 1Institute of Molecular Physiology and Genetics, Centre of Biosciences, Slovak Academy of Sciences, Dúbravská cesta 9, 84005 Bratislava, Slovakia; kristina.simonicova@savba.sk (K.Š.); lubos.janotka@savba.sk (L.J.); helena.kavcova@savba.sk (H.K.); zdena.sulova@savba.sk (Z.S.); 2Department of Biology, Faculty of Medicine and Dentistry, Palacky University Olomouc, Hnevotinska 3, 77515 Olomouc, Czech Republic; 3Institute of Biochemistry and Microbiology, Faculty of Chemical and Food Technology, Slovak University of Technology in Bratislava, Radlinského 9, 81237 Bratislava, Slovakia

**Keywords:** myelodysplastic neoplasms (syndromes), acute myeloid leukemia, resistance, 5-azacytidine, uridine-cytidine kinase, pyrimidine synthesis, teriflunomide

## Abstract

**Simple Summary:**

Although significant progress has been made in the treatment of myeloid malignancies, this issue remains a focus of interest for a large number of research teams. Current research is especially focused on elderly patients who are not suitable for intensive chemotherapy and bone marrow transplantation or patients who have achieved remission after such therapy but subsequently enter into relapse of the disease. Here, therapy with demethylating agents is indicated. However, we do not know of another treatment option in the case of resistance to such treatment. Therefore, knowledge of the causes and mechanisms of resistance to demethylating agents is an essential issue for improving the treatment of such patients.

**Abstract:**

Three AML cell variants (M/A, M/A* from MOLM-13 and S/A from SKM-1) were established for resistance by the same protocol using 5-azacytidine (AZA) as a selection agent. These AZA-resistant variants differ in their responses to other cytosine nucleoside analogs, including 5-aza-2′-deoxycytidine (DAC), as well as in some molecular features. Differences in global DNA methylation, protein levels of DNA methyltransferases, and phosphorylation of histone H2AX were observed in response to AZA and DAC treatment in these cell variants. This could be due to changes in the expression of uridine-cytidine kinases 1 and 2 (UCK1 and UCK2) demonstrated in our cell variants. In the M/A variant that retained sensitivity to DAC, we detected a homozygous point mutation in UCK2 resulting in an amino acid substitution (L220R) that is likely responsible for AZA resistance. Cells administered AZA treatment can switch to de novo synthesis of pyrimidine nucleotides, which could be blocked by inhibition of dihydroorotate dehydrogenase by teriflunomide (TFN). This is shown by the synergistic effect of AZA and TFN in those variants that were cross-resistant to DAC and did not have a mutation in UCK2.

## 1. Introduction

Myelodysplastic neoplasms (previously known as myelodysplastic syndromes, MDS) and acute myeloid leukemia (AML) share a common denominator in abnormal myeloid hematopoiesis, which leads to cytopenias and an increased presence of immature myeloid blasts in the bone marrow and peripheral blood. One of the treatment options for these diseases is the use of the hypomethylating agents (HMAs) 5-azacytidine (AZA) and 5-aza-2′-deoxycytidine (DAC) [1,2].

AZA, commercially available under the name Vidaza, has been approved by the European Medicines Agency (EMA) for the treatment of patients with malignancies caused by defective differentiation in the myeloid branch of hematopoiesis who are not suitable for hematopoietic stem cell transplantation. The EMA has recommended the use of Vidaza for patients with MDS, chronic myelomonocytic leukemia (CMML), AML with 20–30% blasts, and multilineage dysplasia and adult patients aged 65 years or older with AML with >30% blasts in bone marrow [3]. In the USA, Vidaza was also approved by the United States Food and Drug Administration (FDA) for the treatment of patients with MDS and CMML [4]. Moreover, in adult patients with AML who achieve first complete remission (CR) or CR with incomplete blood count recovery following intensive induction chemotherapy and are not able to complete intensive curative therapy, an oral form of AZA, commercially available under the name Onureg, is indicated for continued treatment by both the FDA and EMA [5,6]. Onureg was proven beneficial in postremission treatment, significantly improving overall survival and prolonging relapse-free survival [7]. 

Unfortunately, approximately half of patients may not respond to AZA treatment from the start (primary resistance), and others may become unresponsive during repeated cycles of HMA therapy (secondary resistance) [8]. For a precise understanding of the ways in which myeloid blasts escape from the therapeutic effects of HMAs, it is necessary to understand in detail the changes in the molecular events that enable this escape. We drew attention to specific molecular mechanisms leading to resistance to HMAs in a previous paper [9]. For a precise molecular analysis of possible metabolic and regulatory pathways associated with resistance to hypomethylating agents, it is advantageous to use established cell lines of myeloid blasts in which resistance to AZA or DAC was induced by passage with stepwise increasing concentrations of the drugs. 

We have previously described the preparation of AZA-resistant MOLM-13 and SKM-1 cell variants by sequential passaging in medium with increasing concentration of AZA. These variants presented overexpression of the multidrug efflux pump P-glycoprotein (P-gp) and exhibited resistance to P-gp relative substances in addition to resistance to AZA [10]. Similarly, AZA-resistant variants of HEL cells (a cell line of leukemic erythroblasts) showed increased levels of P-gp expression [11]. Such cells exhibit a multidrug resistance of the P-glycoprotein type and may not have activated specific mechanisms typical for the development of resistance to demethylating agents. Therefore, in a recent paper, we established AZA- and DAC-resistant variants of MOLM-13 cells with continuous control for the negativity of P-gp expression [12]. In the presented paper, we describe the properties of the original variant MOLM-13/AZA (without cross-resistance to DAC) [12], the newly prepared variant MOLM-13/AZA* (also resistant to DAC), and the new variant SKM-1/AZA (with decreased sensitivity to DAC) using the same induction protocol controlling the negativity of P-gp expression with the aim of identifying molecular features responsible for HMA resistance.

Studying possible mechanisms of resistance, we mainly focused on the uptake of HMAs and their fate in sensitive and resistant cells. Both AZA and DAC are prodrugs that must be phosphorylated to the corresponding triphosphates in order to be incorporated into DNA or RNA and exert their effects [13]. Since HMAs are cytidine and deoxycytidine analogs, they use enzymes from the pyrimidine salvage pathway for their activation (Figure 1, red and blue). This pathway is often used by cancer cells to synthesize pyrimidine nucleotides required for DNA replication and mRNA synthesis. Although this pathway is more energy efficient, cells can also use the de novo pyrimidine synthesis pathway to maintain CTP and dCTP pools (Figure 1, green) [14]. For this reason, we also focused on the potential use of de novo pyrimidine synthesis inhibition in cells with HMA resistance.

## 2. Materials and Methods

### 2.1. Cell Culture Conditions

Two cell lines were used in this study. The MOLM-13 cell line (ACC 554) and the SKM-1 cell line (ACC 547) were derived from the peripheral blood of a 20-year-old and a 76-year-old man with AML following MDS, respectively (both supplied by Leibniz-Institute DSMZ-Deutsche Samsung von Microorganism und Zellkulturen GmbH, Braunschweig, Germany). The sensitive MOLM-13 and SKM-1 cell lines were adapted to 5-azacytidine (AZA, Sigma Aldrich, St. Louis, MO, USA) over a 6-month period with repeated passaging in medium containing AZA in stepwise increasing concentrations beginning at 0.1 nM up to final concentration of 1 μM. This procedure yielded AZA-resistant SKM-1/AZA and MOLM-13/AZA* cell variants. The MOLM-13/AZA cell variant was described previously [12]. The cell lines were cultured in RPMI 1640 medium with L-glutamine containing 12% fetal bovine serum (both from Gibco, Langley, OK, USA), 100,000 units/L penicillin, and 50 mg/L streptomycin (both from Sigma Aldrich, St. Louis, MO, USA) at 37 °C in a humidified atmosphere containing 5% CO_2_.

### 2.2. Determination of the Number and Viability of Cells

Sensitive and resistant MOLM-13 and SKM-1 cells were incubated under standard culture conditions with various concentrations of AZA, DAC, teriflunomide (TFN, Sigma Aldrich, St. Louis, MO, USA), or combinations of these drugs for 72 h. The cell lines were treated with AZA/DAC/TFN every 24 h. The number and viability of cells were determined by measuring the plasma membrane integrity of individual cells through changes in electrical resistance induced by cells passing through the detector in the CASY Model TT Cell Counter (Roche Applied Sciences, Madison, WI, USA) according to the manufacturer’s protocol.

### 2.3. MTS Assay and Determination of IC_50_ Values

Sensitive and resistant MOLM-13 and SKM-1 cells were incubated under standard culture conditions with different concentrations of cytarabine or gemcitabine for 48 h and various concentrations of AZA, DAC, TFN, or their combinations for 72 h. The cell lines were treated with AZA/DAC/TFN every 24 h, and cytarabine and gemcitabine were added only once. After cultivation, the CellTiter 96^®^ AQueous One Solution cell proliferation assay (MTS assay, Promega, Madison, WI, USA) was used to determine the metabolic activity of cells according to the manufacturer’s protocol. The IC_50_ (half maximal inhibitory concentration) was computed by nonlinear regression according to Equation (1) using SigmaPlot 8.02 software (Systat Software. Inc., San Jose, CA, USA):(1)$$N=a+A\times \exp\left[\ln\left(0.5\right)\times {\left(\frac{c}{{\mathrm{IC}}_{50}}\right)}^{n}\right]$$
where N in % is the metabolic activity of cells after drug treatment at concentration c; a + A in % is the metabolic activity of control/untreated cells; A represents the metabolic activity that is suppressed by the respective drug; IC_50_ is the half maximal inhibitory concentration; and n represents order exponents for cytotoxic effects. The data represent computed values ± standard error with 30 degrees of freedom.

### 2.4. Determination of mRNA Gene Expression

Total RNA was isolated using TRI Reagent (MRC, Cincinnati, OH, USA) according to the manufacturer’s instructions. Reverse transcription (RT) was performed with 1 μg of RNA using a RevertAid™ H Minus First-Strand cDNA synthesis kit (Thermo Fisher Scientific, Waltham, MA, USA) according to the manufacturer’s protocol. PCR was performed in a total volume of 25 μL using a DreamTaq PCR kit (Thermo Fisher Scientific, Waltham, MA, USA) according to the manufacturer’s protocol. After heating the samples at 95 °C for 5 min, samples were subjected to 30 cycles of denaturation (95 °C, 30 s), annealing (temperatures in Table 1, 30 s), extension (72 °C, 90 s), and a final extension (72 °C for 10 min). The PCR products were separated on a 1.5% agarose gel (Lonza, Rockland, ME, USA), and the gel was visualized with GelRed™ nucleic acid gel stain (Biotium, Fremont, CA, USA) using an Amersham Imager 600 (GE Healthcare Europe GmbH, Pittsburgh, PA, USA). The expression of genes at the mRNA level was quantified by densitometric analysis of PCR product bands using ImageQuant software (GE Healthcare Europe GmbH, Pittsburgh, PA, USA) and normalized to *ACTB*.

### 2.5. Sequence Analysis

For sequence analysis, RNA isolation and RT were performed as in 2.4, but Pfu polymerase was used in the PCR. PCR was performed in a total volume of 20 μL using a Phusion™ High-Fidelity DNA Polymerase PCR kit (Thermo Fisher Scientific, Waltham, MA, USA) according to the manufacturer’s protocol. PCR thermal cycling conditions were as follows: initial denaturation (98 °C, 5 min); 30 cycles of denaturation (98 °C, 1 min); annealing (temperature in Table 1, primers marked with *, 30 s); and extension (72 °C, 2 min), followed by final extension (72 °C, 10 min). After separation on a 1.5% agarose gel, the PCR products were extracted from the gel with a GeneJET Gel Extraction Kit (Thermo Fisher Scientific, Waltham, MA, USA) according to the manufacturer’s protocol. The sequences of the PCR products were determined by Sanger sequencing (Eurofins Genomics Germany GmbH, Ebersberg, Germany). 

### 2.6. Detection of UCK1, UCK2, DCK, γ-H2AX, α-TUB and GAPDH Protein Levels

For γ-H2AX detection, cells were cultured in standard medium for 24 h with or without 0.25 and 0.5 μM AZA or DAC and 72 h with or without 1 μM AZA or DAC. The cells were treated with AZA/DAC every 24 h. After incubation, the cells were harvested, and proteins were extracted by RIPA lysis buffer containing 50 mM Tris-Cl (pH 8.0), 1% Triton X-100, 0.5% sodium deoxycholate, 0.1% SDS, 150 mM NaCl, and protease inhibitor cocktail from Sigma-Aldrich (Saint Louis, MO, USA). The protein concentrations were measured by a Pierce™ BCA Protein Assay kit (Thermo Fisher Scientific, Waltham, MA, USA). The protein samples were separated by sodium dodecyl sulfate-polyacrylamide electrophoresis (SDS–PAGE) in a 12% gel. The proteins were then transferred by electroblotting to a nitrocellulose membrane (GE Healthcare Europe GmbH, Vienna, Austria). The primary antibodies used were UCK1 (HPA050969) and UCK2 (SAB1411384) (both from Sigma-Aldrich, St. Louis, MO, USA); DCK (ab96599) from Abcam, Cambridge, UK; GAPDH (MAB374) (EMD Millipore Chemicals, Billerica, USA); and γ-H2AX (CST 9718) and α-tubulin (CST 3873) (both from Cell Signaling Technology, Danvers, MA, USA). Goat anti-rabbit (SC-2054) and mouse anti-rabbit antibodies (SC-2357) (both from Santa Cruz Biotechnology, Dallas, TX, USA) and horse anti-mouse (CST 7076) and goat anti-rabbit antibodies (CST 7074) (both from Cell Signaling Technology, Danvers, MA, USA), all conjugated with horseradish peroxidase, served as secondary antibodies. Protein bands were visualized by ECL detection (GE Healthcare Europe GmbH, Vienna, Austria) and Amersham Imager 600 (GE Healthcare Europe GmbH, Pittsburgh, PA, USA). Protein quantities were established by densitometry using ImageQuant software (GE Healthcare Europe GmbH, Pittsburgh, PA, USA) and normalized to GAPDH or α-Tubulin.

### 2.7. Detection of Cell Death Mode 

Cells were incubated with various concentrations of AZA, DAC, TFN, or a combination of these drugs for 24 or 72 h under standard culture conditions. The cells were treated with AZA/DAC/TFN every 24 h. After the incubation period, the proportions of apoptotic and necrotic cells were measured using an Annexin V (Roche, Mannheim, Germany)/propidium iodide (Calbiochem, San Diego, CA, USA) assay. The cells were washed with PBS and gently resuspended in binding buffer containing 0.5 μg/mL FITC-labeled Annexin V. The mixtures were incubated for 15 min at room temperature in the dark. Finally, propidium iodide (final concentration of 0.6 μg/mL) was added to each sample, after which the samples were analyzed by flow cytometry using an Accuri C6 flow cytometer (BD Bioscience, San Jose, CA, USA).

### 2.8. Global DNA Methylation Status Determination 

Cells were incubated with or without 0.5 μM AZA or DAC for 72 h. The cells were treated with AZA/DAC every 24 h. Total DNA was isolated using TRI Reagent (MRC, Cincinnati, OH, USA) according to the manufacturer’s instructions. Global DNA methylation status was determined by a Methylated DNA Quantification Kit (Abcam, Cambridge, UK) according to the manufacturer’s instructions. 

### 2.9. Statistical Analysis and Data Processing 

Numerical data are expressed as the mean ± SD of three independent measurements. Statistical significance was assessed by unpaired Student’s t test. Correlations were determined using Pearson correlation analysis. SigmaPlot 8.0 software (Systat Software, Inc., San Jose, CA, USA) was used. 

## 3. Results

### 3.1. Resistance to AZA and Cross-Resistance to DAC and Other Deoxycytidine Analogs

Three AZA-resistant cell sublines were prepared in our laboratory: one from the SKM-1 cell line (SKM-1/AZA—S/A) and two from the MOLM-13 cell line (MOLM-13/AZA—M/A and MOLM-13/AZA*—M/A*). The M/A subline was already described in our previous work [12]. Using the MTS assay, we determined the half maximal inhibitory concentration (IC_50_) values of four (deoxy)cytidine analogs—AZA, DAC, gemcitabine (GEM), and cytarabine (AraC)—in our cell variants. M/A did not show any cross-resistance to other analogs. However, S/A showed a slightly lower sensitivity to the other three analogs, while M/A* showed considerably increased cross-resistance to the analogs, especially DAC (Figure 2). Even at a concentration of 40 μM, DAC did not show a half-maximal inhibitory effect on the M/A* cells; however, a significant inhibitory effect was already observed at a concentration of 1 μM (Appendix A).

After incorporation into DNA, HMAs exert two main effects: DNA hypomethylation by DNA methyltransferase (DNMT) inhibition and DNA damage response induction [16]. Therefore, to analyze why HMAs are not able to exert their cytotoxic effect on the resistant variants, global DNA methylation status and the level of histone H2AX phosphorylation (γ-H2AX, marker of DNA damage) were assessed. Using a methylated DNA quantification kit, we determined the global DNA methylation status of our cell variants. We did not observe significant changes in DNA methylation status between parental and AZA-resistant cell variants. Both drugs caused a decrease in DNA methylation in both parental cell lines. AZA did not induce a significant reduction in DNA methylation in any of the three AZA-resistant variants; however, DAC significantly reduced DNA methylation in S/A and M/A, but not in M/A* (Figure 3A). Levels of γ-H2AX were measured first after 24 h of cultivation with 0.25 and 0.5 μM AZA/DAC treatment (Figure 3B and Appendix A). Especially in parental cell lines, we detected some increase in H2AX phosphorylation even after treatment with these low concentrations, although we did not detect induction of apoptosis at this timepoint with 0.5 μM HMAs (Appendix A). Next, we analyzed the levels of γ-H2AX after 72 h of treatment with 1 μM AZA/DAC. While there was no increase in γ-H2AX in MOLM-13/AZA* cells after treatment with either HMA, increased levels of γ-H2AX were observed in both MOLM-13/AZA and SKM-1/AZA cells after both AZA and DAC treatment (Figure 3B and Appendix A).

### 3.2. Expression of Genes Involved in Metabolism and Transport of the Hypomethylating Agents

Using RT–PCR, we measured the mRNA expression of genes involved in the metabolism and transport of AZA and DAC (Figure 4A and Appendix A). For most of the genes, we did not find significant differences in expression between sensitive cell lines and their AZA-resistant counterparts. We observed slightly increased expression of cytidine deaminase (*CDA*) in M/A and M/A* cells compared to MOLM-13 cells. In M/A, we also observed slightly decreased expression of dihydroorotate dehydrogenase (*DHODH*) (Figure 4A and Appendix A). We also measured the expression of *SLC28A1* and *SLC28A3* (Solute Carrier Family 28 Member 1 and 3, respectively), but neither was expressed in any of our cell variants.

For both hypomethylating agents, phosphorylation to the corresponding monophosphate form is the rate limiting step in their activation [17]. Therefore, we also studied the expression of UCK1, UCK2 (uridine-cytidine kinase 1 and 2, respectively), and DCK (deoxycytidine kinase) at the protein level. DCK protein was present in all AZA-resistant cell variants despite the cross-resistance to DAC observed in S/A and M/A*; however, in M/A*, protein expression of DCK was significantly downregulated (Figure 4B,C). The PCR products of the DCK whole coding regions in these two cell variants were sequenced, and no mutations were observed. UCK1 protein was downregulated in both S/A and M/A*. In M/A*, we also observed the downregulation of UCK2. This protein was also downregulated in M/A (Figure 4B,C).

The RT–PCR products of whole *UCK1* and *UCK2* protein coding regions were sequenced. We found a homozygous point mutation of *UCK2* in the M/A cell variant (Figure 5A) and a heterozygous point mutation of *UCK1* in the S/A cell variant (Figure 5B), both resulting in amino acid substitutions (L220R and R168G, respectively).

### 3.3. Impact of the DHODH Inhibitor Teriflunomide on AZA-Resistant Cells

Teriflunomide (TFN), an inhibitor of dihydroorotate dehydrogenase (DHODH), exerted slightly different impacts on our three AZA-resistant cell sublines. In the case of M/A, TFN caused a decrease in viable cell counts compared to the untreated control, but it did not induce cell death even at the highest concentration used. In M/A*, the number of viable cells was also lower than in the control, but we also observed an increased number of apoptotic cells in the population. In S/A, we observed only a small decrease in the metabolic activity of the cells, with no change in their number. We also did not observe a significant increase in the apoptotic cell counts. When compared to sensitive parental cell lines, TFN had a similar or weaker effect on AZA-resistant cell variants (Figure 6). Except for S/A, the values of metabolic activity strongly correlated with the values of the viable cell number, so the metabolic activity reflects the number of viable cells (Appendix A). In the case of S/A, it seems that TFN induces a slightly greater effect on the metabolic activity of the cells than on their counts.

Next, we measured the impact of TFN and AZA combined treatment on our AZA-resistant cell variants (Figure 7 and Figure 8). We cultured cells in medium with or without 5 μM TFN and with AZA at three concentrations (1, 5, and 10 μM) for 72 h. Both drugs were added to the medium every 24 h. In M/A* and S/A cell variants, we observed AZA-induced apoptosis in cells cotreated with TFN (Figure 8). TFN alone, as well as AZA alone (except for a 10 μM concentration of AZA for the S/A cell variant), exerted only minor impacts on cell viability, but together, the drugs achieved a significant synergistic effect. A synergistic effect can be expected when the response to the treatment with a combination of two drugs (Figure 7, gray bars) is higher than the summed effects of the drugs when used individually (Figure 7, gray line). However, in M/A, we did not observe this synergistic effect, and AZA did not induce apoptosis when used in combination with TFN (Figure 7 and Figure 8).

We also measured the impacts of DAC and TFN combined treatment in the two cell variants that showed cross-resistance to DAC. We used 5 μM TFN as in the previous experiments and DAC at four concentrations (0.5, 1, 2, and 5 μM). However, in this case, we did not observe a significant synergistic effect (Figure 9).

## 4. Discussion

In our previous paper, we introduced our AZA-resistant cell line MOLM-13/AZA derived from the human AML cell line MOLM-13, which did not show cross-resistance to DAC [12]. In this study, we developed another AZA-resistant subline, SKM-1/AZA, which was derived from another human AML cell line, SKM-1. In this cell variant, a slightly lower sensitivity toward DAC than in the parental cell line was observed. Moreover, we also developed another AZA-resistant cell line from MOLM-13—MOLM-13/AZA*, which is considerably less sensitive to DAC than the parental cell line (Figure 2). This novel cell variant is resistant to both AZA and DAC, but with different dose–response curves (Appendix A). DAC causes a slight decrease in the viability of these cells at concentrations as low as 1 µM; however, even 40 µM DAC does not cause a 50% decrease in viability. Here, then, the IC_50_ value cannot be calculated but is higher than 40 µM (Appendix A). In contrast, AZA has an IC_50_ value of approximately 40 µM, although viability only began to decline at higher concentrations than with DAC (Appendix A). The differences between our AML cell variants with AZA-induced resistance in the development of cross-resistance to DAC are consistent with previous studies conducted by different groups. In most cell variants, induction with AZA led to at least a partial reduction in sensitivity to DAC as well [18,19,20,21,22], but in some cases, cell variants that developed resistance to AZA remained sensitive to the second HMA [22,23,24]. Interestingly, Murase et al. (2016) prepared AZA-resistant cell variants from the same cell lines (MOLM-13 and SKM-1) as we did. Both of their AZA-resistant cell variants showed a decreased sensitivity toward DAC. Similar to our results, one of the cell variants showed only a slight decrease in sensitivity to this drug, while the second variant showed an even lower sensitivity to DAC than to AZA. However, a more DAC-cross-resistant variant was established from the SKM-1 cell line, in contrast to our cell variant MOLM-13/AZA* [19]. Taken together, these results may suggest that the mechanism of resistance to HMA may not always depend on the cell line type or induction protocol alone. The results of studies on AZA-resistant cell variants showing a higher proportion of cells cross-resistant to DAC are also supported by a few clinical studies in which patients were treated with DAC after AZA treatment failure. In two of these studies, none of the patients responded to DAC treatment [25,26], while in three others, the response rate ranged from 19% to 28% [27,28,29]. 

In addition to the two HMAs, we determined cytotoxic effects for two other 2-deoxycytidine analogs, gemcitabine and cytarabine, with modifications in the furanose skeletons of these deoxyribonucleotides that do not have demethylating effects. We also calculated the IC_50_ values for these two substances (Figure 2). Compared to the parental cell lines, we observed a slight decrease in sensitivity to these two analogs in those cell variants that exhibited a lower sensitivity to DAC. Examining the cross-resistance of AZA-resistant cell sublines to gemcitabine and cytarabine, Murase et al. (2016) also observed a slight reduction in the sensitivity to these analogs in variants with a reduced resistance to DAC [19].

A more pronounced effect of DAC than AZA was observed in both our parental cell lines (Figure 2). Consistent with this, both the EMA and FDA recommend the application of AZA in a higher dosage (75 mg/m^2^ for 7 days) than DAC (20 mg/m^2^ for 5 days). However, both AZA and DAC have demonstrated efficacy in the treatment of MDS and AML that preceded the drugs’ approval [3,4,30,31]. Therefore, both drugs should be the subject of intensive future research in the treatment of cancer.

Interestingly, IC50 of GEM is much lower compared to all the other drugs tested in our cell variants, even in those showing resistance to both AZA and DAC (Figure 2). Drenberg et al. (2019) also observed broad anti-leukemic activity of GEM in several leukemic cell lines and found it to be more effective than AraC, standardly used in AML treatment, both in vitro and in vivo [32]. The reason for the greater effect of GEM on leukemic cells compared to other pyrimidine analogs could lie in its ability of self-potentiation and several mechanisms of action proposed for the drug; however, the relevance of mechanisms other than incorporation of GEM triphosphate into DNA are in question [13]. A Phase II study of GEM in children with relapse/refractory acute leukemias did not demonstrate significant activity at an evaluated dose and schedule; however, a large number of patients included in this trial were previously exposed to many other antineoplastic drugs [33]. We are not aware of any single-agent clinical study of GEM conducted in adult AML patients so far, but it might be worth trying. However, apart from the efficacy, the toxicity of the treatment remains an important issue, especially in elderly patients.

To determine the mechanism of resistance to AZA and 2-deoxycytidine analogs in our cell variants, we focused primarily on the changes in the expression of genes involved in the entry of these analogs into cells and their subsequent activation. Since HMAs must first enter cells by transport across the plasma membrane, we measured the gene expression (at the mRNA level) of four nucleoside transporters capable of transporting pyrimidine nucleosides and their analogs [9]. The *SLC28A1* and *SLC28A3* genes encoding Na^+^-mediated concentrative nucleoside transporters hCNT1 and hCNT3 were not expressed in any of our cell variants (not shown). Very low or no expression of these two genes was reported in leukemia and solid tumor cell lines, as well as in blasts from AML patients [15,34,35]. Despite the reported decrease in AZA sensitivity after hENT1 inhibition, changes in the expression of hENT1 or other nucleoside transporters do not seem to be the cause of acquired AZA resistance in cell lines [9]. None of our three AZA-resistant cell variants showed a decreased expression of *SLC29A1* or *SLC29A2* encoding equilibrative nucleotide transporters hENT1 and hENT2, respectively (Figure 4A). Similarly, no remarkable changes were observed in the expression of these genes in AZA-resistant cell variants compared to the parental cell lines in two different studies [18,36]. All of these facts suggest that the reduced sensitivity of our cell variants to HMA and other 2-deoxycytidine analogs, in which resistance was induced by AZA, does not appear to be caused by any changes in the entry of cytidine analogs into the cell.

HMAs, as well as GEM and AraC, are prodrugs, and to be incorporated into nucleic acids, they must first be enzymatically activated by sequential phosphorylation to mono-, di-, and triphosphate forms. We therefore measured the mRNA expression of UCK1 and UCK2, which are capable of cytidine (AZA) and uridine phosphorylation, as well as DCK, which is responsible for deoxycytidine (DAC, GEM, and AraC) phosphorylation to the corresponding monophosphate forms [13]. While DAC is incorporated into DNA exclusively, AZA is primarily incorporated into RNA, and only approximately 20% of AZA is converted to the deoxyribonucleotide form by reduction of the respective diphosphate, which is catalyzed by ribonucleotide reductase (RNR) [37]. This enzyme consists of two subunits, the large subunit RRM1 and the small subunit existing in the two isoforms RRM2 and RRM2B [38]. We did not observe any differences in their expression at the mRNA level (Figure 4A). We also decided to measure the expression of genes encoding the 5′-nucleotidase (encoded by *NT5C3A*) that catalyzes the dephosphorylation of pyrimidine 5’-monophosphates, as well as deoxynucleoside triphosphate triphosphohydrolase 1 (*SAMHD1*). Oellerich et al. (2019) observed increased SAMHD1 expression at both the mRNA and protein levels in AZA-resistant cell variants developed from the HL-60 cell line, which did not express this gene. Degradation of SAMHD1 in the AZA-resistant cell subline sensitized it to DAC [20]. However, both of our parental cell lines (SKM-1 and MOLM-13) already expressed SAMHD1 at the mRNA level, and in AZA-resistant sublines, no further increase in expression was observed (Figure 4A). 

The only change in the gene expression of enzymes involved in the metabolism of HMAs that we observed at the mRNA level when comparing our resistant and sensitive cell variants was the expression of CDA, which was upregulated in M/A and M/A* (Figure 4A). Similar results were observed at the protein level by Oellerich et al. (2019) in three AZA-resistant cell variants derived from the human AML cell line HL-60 [20]. However, it is questionable whether AZA deamination by CDA is truly an undesirable phenomenon, as the product of AZA deamination, 5-azauridine, might be phosphorylated and then incorporated into RNA, which could induce effects such as protein synthesis attenuation [9]. Moreover, we observed upregulation of CDA in the non-cross-resistant cell variant and in one of the cell variants cross-resistant to DAC, but not in the other one, even though both HMAs are considered to be substrates of this enzyme [39]. 

Despite the lack of changes in the expression of genes at the mRNA level, we observed many changes at the protein level in the expression of UCK1, UCK2, and DCK (Figure 4). We decided to analyze these particular proteins, considering the results of the global DNA methylation status and γ-H2AX expression level determination suggesting differences in AZA and DAC DNA incorporation rates (Figure 3). Both HMAs exert two main effects after incorporation into DNA. One is DNA hypomethylation by DNMT inhibition, and the second is the induction of the DNA damage response [16]. The phosphorylation of H2AX is a marker of DNA damage and repair. The formation of γ-H2AX is considered to be an early event in the DNA damage response not only to DNA double-strand breaks, but also to covalent DNA adduct formation [40]. 

Consistent with the absence of cross-resistance of the M/A cell variant to DAC (Figure 2), DCK protein expression did not differ from that of the parental cell line (Figure 4B,C). In M/A, DAC caused global DNA hypomethylation and upregulation of H2AX phosphorylation (Figure 3). However, UCK2 was significantly downregulated in this cell variant (Figure 4B,C), even though downregulation was not detected at the mRNA level (Figure 4A). The protein downregulation might be related to a novel single nucleotide mutation found in exon 7 (Figure 5A). This missense mutation leads to an amino acid substitution from leucine to arginine. This amino acid is localized in the center of the helix, and the amino acid substitution is predicted to lead to structural alteration in this motif. In addition, this helix is located relatively close to the ATP binding site, which could have an effect on UCK2 activity. The mutation is predicted to be deleterious according to MutationTaster2021 [41] and affect protein function with a score of 0.00 according to the SIFT web server [42]. Thus, although the UCK2 protein is present in M/A cells, it may have reduced function. 

Unlike DAC, AZA does not induce global DNA demethylation in M/A cells; however, the slight upregulation of γ-H2AX (Figure 3) may indicate that some AZA is still phosphorylated in these cells, presumably by UCK1, whose expression is not lower than that in the parental cell line (Figure 4B,C). This reduced fraction of AZA, which is subject to phosphorylation, may then elicit the abovementioned mild responses upon incorporation into nucleic acids.

In M/A*, UCK1, UCK2, and DCK were all downregulated compared to the parental cell line at the protein level (Figure 4B,C). Neither DNA hypomethylation nor H2AX phosphorylation were considerably increased (Figure 3). Similarly, in the study by Hur et al. (2017), neither AZA nor DAC caused downregulation of DNMT1 protein expression in AZA-resistant cell variants cross-resistant to DAC [18].

However, in the case of S/A, despite a partially reduced sensitivity to DAC, DCK protein was not downregulated (Figure 4B,C), and treatment with DAC led to global DNA hypomethylation and DNA damage induction, but some of these effects were weaker or delayed compared to the parental cell line (Figure 3). Consistently, AZA-resistant cell variants cross-resistant to DAC showed an unchanged expression of DCK protein compared to parental cell lines in a study by Murase et al. (2016) [19]. In our study, we checked the sequence of the whole DCK coding region, and we did not observe any mutations in S/A or M/A*. This excludes any reason to hypothesize that DCK might be dysfunctional. The level of UCK2 protein in S/A cells was the same as that in the parental cell line, and the level of UCK1 was slightly lower. We also found a mutation of UCK1 in the S/A cell variant (Figure 5B). According to MutationTaster2021, this mutation is predicted to be deleterious and to cause the loss of helix [41]. Additionally, the SIFT web server predicts the substitution R168G to affect protein function with a score of 0.00 [42]. However, this mutation is heterozygous, so the functional protein can still be expressed from the unmutated allele.

In other studies, UCK2 was downregulated in AZA-resistant AML cell variants at the mRNA level [21] or at the protein level in AZA-resistant cells from histiocytic lymphoma, AML, and adult T-cell leukemia-lymphoma [19,22]. In the latter, the complete absence of protein was the consequence of a splice donor site mutation [22]. The downregulation of UCK2 at the mRNA level was also observed in MDS patients with AZA treatment relapse [43]. In two different studies, the downregulation of UCK2 at the protein level was not observed in AZA-resistant AML cells [20,36]; however, in one of the studies, UCK2 mutations were observed in these cells, which were attributed to possible enzyme inactivation [36].

Considering the changes in AZA-resistant cell sublines related to the phosphorylation of AZA catalyzed by enzymes important in the pyrimidine salvage pathway, we were interested in the question of whether inhibition of the de novo pyrimidine synthesis pathway would have different effects on AZA-resistant cell variants compared to the parental cell lines. We expected the AZA-resistant cells to be more sensitive to the inhibition of DHODH by TFN (drug approved by the FDA and EMA for treatment of relapsing forms of multiple sclerosis [44,45]), as was observed in the study by Imanishi et al. (2017) with respect to the metabolic activity/viability of the cells [46]. However, we did not observe a trend of higher sensitivity of AZA-resistant cells to this drug. At most of the concentrations, the effect of TFN on metabolic activity was the same or even weaker in the AZA-resistant cells than in the parental cells, except for the 3 μM concentration in the M/A cell subline (Figure 6B). Moreover, comparing the relative number of viable cells and apoptotic cell proportions, we observed an even weaker effect of TFN on AZA-resistant cells than on parental cells (Figure 6A,C).

It is also interesting that TFN seems to have slightly different effects on each of our AZA-resistant cell variants. In M/A*, similar to the parental cell lines, the effect of TFN on the number of viable cells correlated with the proportions of viable and apoptotic cells. However, the decrease in viable cell number in the M/A group was not reflected in the proportions of viable and apoptotic cells. It seems that in this case, TFN treatment caused cytostatic rather than cytotoxic effects on the cells. Finally, in S/A, TFN does not seem to have an effect on viable cell counts or on a significant increase in the proportion of apoptotic cells. In this cell variant, we observed an effect only on metabolic activity (Figure 6 and Appendix A). 

We also studied the effect of AZA and TFN combination treatment on AZA-resistant cells. In S/A and M/A* cell variants, a synergistic effect of the drugs was observed. The presence of TFN potentiated AZA-induced apoptosis (Figure 7 and Figure 8). This effect was also observed in the study by Imanishi et al. (2017), in which they used the combination of the same drugs to treat two different human leukemic cell lines (U937 and HL-60) [46]. Similarly, in their previous study, the combination treatment of AZA with the cytidine triphosphate synthetase inhibitor 3-deazauridine led to AZA-induced growth inhibition [21]. It is interesting that our two cell sublines in which the AZA and TFN cotreatment was effective and both AZA-resistant cell sublines from Imanishi et al. (2014 and 2017) showed reduced sensitivity to DAC [21,46]. However, the treatment of S/A and M/A* with a combination of TFN and DAC did not lead to DAC-induced apoptosis, and we did not observe a synergistic effect of these drugs (Figure 9). Our third AZA-resistant cell variant, M/A, which is not cross-resistant to DAC, did not show any increase in response to AZA when administered to cells in combination with TFN (Figure 7 and Figure 8). 

The differences in the response of our AZA-resistant sublines to the combination of TFN and AZA could be related to the expression and activity of the UCK2 protein. Little knowledge exists regarding UCK1 and UCK2 substrate specificity and affinity to AZA compared to naturally occurring cytidine and uridine. However, a genome-wide CRISPR/Cas9 knockout screen identified only UCK2 (and not UCK1) as a nonredundant and rate-limiting enzyme for AZA activation [47]. Some cytidine analogs were reported to be more efficiently (or even solely) phosphorylated by UCK2 than by UCK1 [48,49,50]. In the study by Van Rompay et al. (2001), phosphorylation of AZA was not proven to be performed by UCK2 or UCK1. However, 6-azacytidine, whose structure strongly resembles the structure of AZA, was phosphorylated by UCK2 more than 10-fold more efficiently than by UCK1 [48]. Human UCK1 and UCK2 share 72% sequence identity [51]. Nevertheless, the catalytic efficiency and affinity to both pyrimidines of UCK2 is considerably higher than that of UCK1 [52]. Thus, it may be expected that AZA will also be preferentially phosphorylated by UCK2. UCK1 is expressed universally in various tissues, while UCK2 is known to be expressed only in human placenta and several types of cancers [51]. Lee et al. (1974) purified UCK from calf thymus. Since the calf thymus is not one of the organs within which UCK2 expression has been described, it can be presumed that the authors isolated UCK1. They determined that the Km values for cytidine and uridine were five- and fourfold lower than the Km for AZA. Cytidine and uridine are potent competitive inhibitors of AZA phosphorylation, whereas AZA is a weak competitive inhibitor of cytidine phosphorylation [53]. Moreover, AZA-CTP was found to be a potent inhibitor of AZA phosphorylation but a weak inhibitor of uridine and cytidine phosphorylation, which may suggest that the feedback inhibition of UCK by AZA-CTP may limit the amount of activated AZA in AZA-treated cells [54].

Taken together, we hypothesize that UCK1 is much less effective in AZA phosphorylation than UCK2. In the cell variants with functional UCK2 (M/A* and S/A), inhibition of de novo pyrimidine synthesis (Figure 1, green) by TFN might force cells to switch to the salvage pathway (Figure 1, red) and incorporate more AZA. However, when using the combination of TFN with DAC, these cells were not forced to phosphorylate DAC (Figure 1, blue) because dCTP for DNA synthesis can be synthesized from cytidine and uridine (Figure 1, red). On the other hand, the lack of AZA-induced apoptosis in the M/A cell variant treated with the TFN and AZA combination might be related to UCK2 mutation and dysfunction. This could render these cells incapable of phosphorylating more AZA and incorporating it into DNA. UCK1 will probably prefer cytidine and uridine and, together with functional DCK, provide enough CTP and dCTP for RNA and DNA synthesis. We know that DCK is active in this cell variant since DAC, AraC, and GEM have very similar effects on cell viability as in the parental cell line (Figure 2).

The variability of the response of our AZA-resistant cell variants to HMAs, specifically highlighting differences between them, is summarized in Table 2.

Since there is still no treatment approved for MDS and AML patients after HMA treatment failure [55], the possibility of including TFN or other de novo pyrimidine synthesis inhibitors in combination with AZA in the treatment protocol should be further investigated and validated. In a specific group of MDS or AML patients with AZA resistance, this combination of drugs could restore the sensitivity of leukemic cells to AZA. Since TFN is already used in treatment of multiple sclerosis patients [44,45], the side effects and off-target toxicity are already known. Therefore, orientation on application of TFN together with AZA in clinical research seems to be reasonable. Moreover, in some preclinical models, some DHODH inhibitors distinct from TFN have been shown to be potentially useful in AML treatment. These inhibitors have been reported to induce both proliferation inhibition and differentiation of AML blasts [56,57,58,59].

## 5. Conclusions

In our laboratory, three AZA-resistant cell variants were prepared using the same induction protocol, but with different observed outcomes. The variability among our cells is in accordance with the diverse results of other research groups. However, despite the different molecular characteristics of our models, specific mutations or changes in the expression of genes involved in the activation of HMAs were observed in all of them. The possible involvement of different pyrimidine nucleotide synthesis pathways in cells seems to play an important role in the response to AZA treatment. 

In AZA-resistant cells showing cross-resistance to DAC, we observed a different response to AZA and TFN combined treatment compared to the cells in which cross-resistance had not developed. After confirmation of these findings in clinical samples, the incorporation of teriflunomide or other substances inhibiting de novo pyrimidine synthesis into the MDS/AML treatment protocol should be considered for a certain group of patients with acquired AZA resistance. 

## Figures and Tables

**Figure 1 cancers-15-03063-f001:**
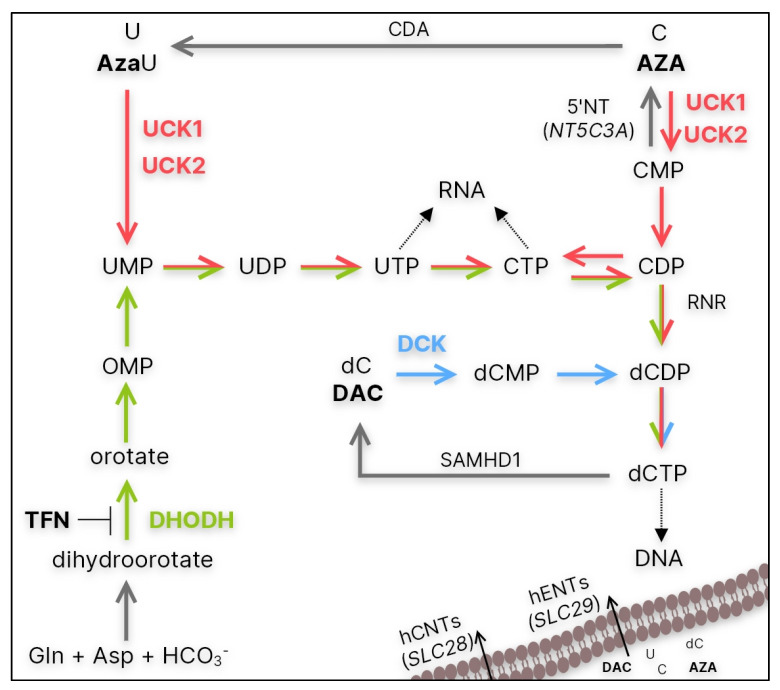
Simplified summary of the de novo (green) and salvage (red and blue) pathways of CTP and dCTP synthesis. CDA—cytidine deaminase; 5′NT—5′-nucleotidase encoded by *NT5C3A*; hENT/hCNT—human equilibrative/concentrative nucleoside transporter encoded by *SLC28* and *SLC29* genes, respectively; UCK1/2—uridine-cytidine kinase 1/2; DCK—deoxycytidine kinase; RNR—ribonucleotide reductase; SAMHD1—Sterile alpha motif and histidine-aspartate domain-containing protein 1; DHODH—dihydoroortate dehydrogenase; AzaU—5-azauridine; TFN—teriflunomide.

**Figure 2 cancers-15-03063-f002:**
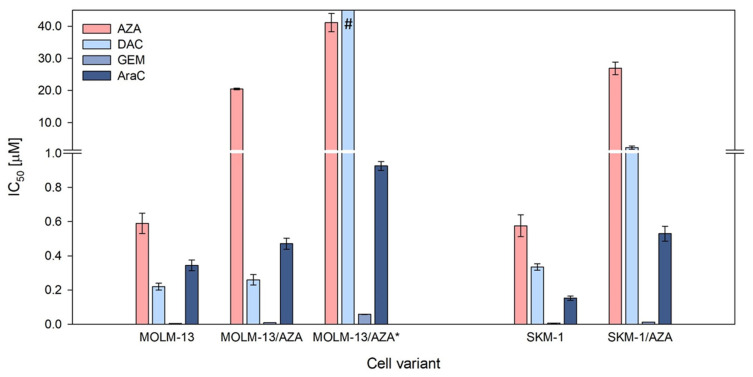
Cell sensitivities to 5-azacytidine (AZA), 5-aza-2′-deoxycytidine (DAC), gemcitabine (GEM), and cytarabine (AraC) expressed as the half maximal inhibitory concentration (IC_50_). The IC_50_ was computed by nonlinear regression according to Equation (1) using SigmaPlot. The data represent computed value ± standard error with 30 degrees of freedom. # = Even at a concentration of 40 μM, DAC did not show a half-maximal inhibitory effect on the cells.

**Figure 3 cancers-15-03063-f003:**
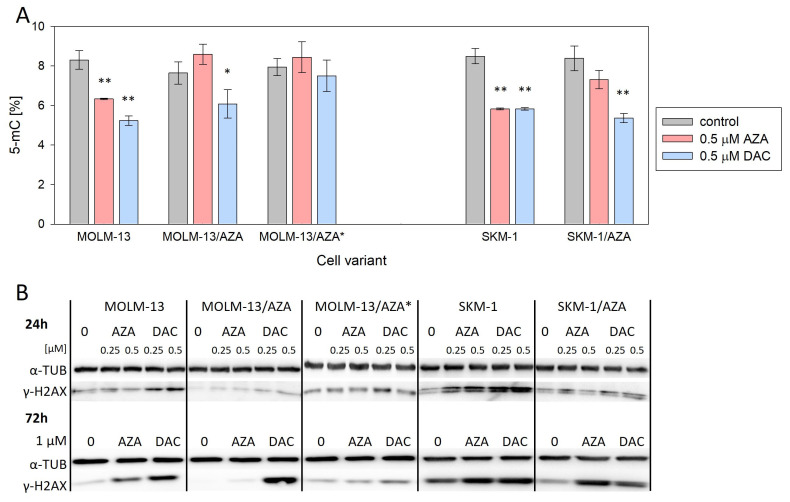
Effect of AZA or DAC on DNA-related events in the cell variants. (**A**) Global DNA methylation status determined by a methylated DNA quantification kit. The cells were incubated for 72 h in the presence or absence of AZA/DAC at a concentration of 0.5 μM. AZA or DAC was added to the cultivation medium every 24 h. Statistical significance is as follows: * *p* ≤ 0.05; ** *p* ≤ 0.01. (**B**) Protein expression of γ-H2AX determined by Western blot analysis. α-TUB was used as an internal control. The cells were incubated for 24 h in the presence or absence of AZA/DAC at 0.25 and 0.5 μM concentrations and 72 h in the presence or absence of AZA/DAC at 1 μM concentration. AZA or DAC was added to the cultivation medium every 24 h.

**Figure 4 cancers-15-03063-f004:**
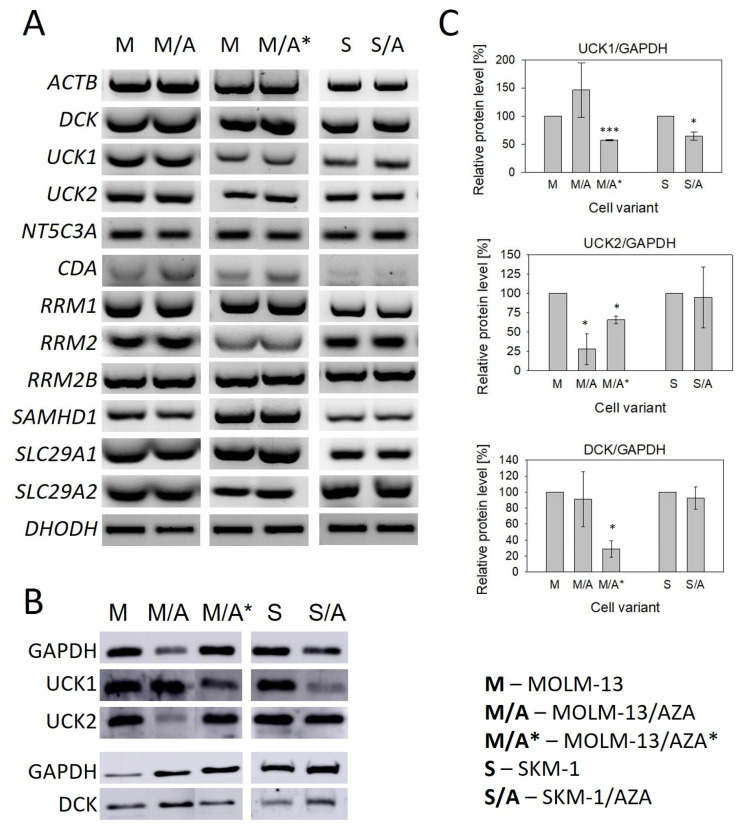
(**A**) Expression of genes involved in the metabolism and transport of cytidine, deoxycytidine, and their analogs AZA and DAC determined by RT–PCR. *ACTB* was used as an internal control. (**B**) Protein expression of UCK1, UCK2, and DCK determined by Western blot analysis. GAPDH was used as an internal control. (**C**) The optical densities of the protein bands were quantified by densitometry and are summarized in the bar plots. The data are expressed as the mean ± SD of three independent measurements. Statistical significance is as follows: * *p* ≤ 0.05; *** *p* ≤ 0.001. *CDA*—cytidine deaminase; *NT5C3A*—gene encoding 5′-nucleotidase; *SLC29A1/SLC29A2*—genes encoding hENT1 and hENT2, respectively; UCK1/2—uridine-cytidine kinase 1/2; DCK—deoxycytidine kinase; *RRM1/RRM2/RRM2B*—genes encoding ribonucleotide reductase subunits; *SAMHD1*—Sterile alpha motif and histidine-aspartate domain-containing protein 1; *DHODH*—dihydroorotate dehydrogenase.

**Figure 5 cancers-15-03063-f005:**
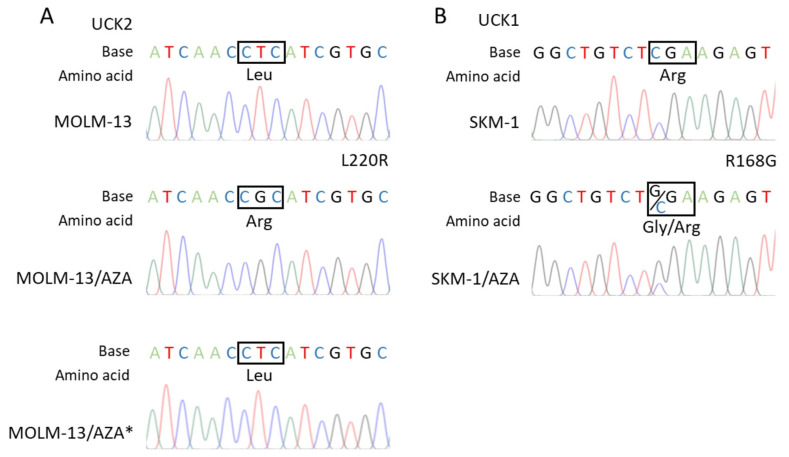
Results of the RT–PCR product sequence analysis of (**A**) the UCK2 coding region in the MOLM-13 parental cell line and its AZA-resistant sublines showing the homozygous mutation site present in the MOLM-13/AZA cell variant but absent in the MOLM-13/AZA* variant. (**B**) UCK1 coding region in the SKM-1 parental cell line and AZA-resistant subline SKM-1/AZA showing the heterozygous mutation site.

**Figure 6 cancers-15-03063-f006:**
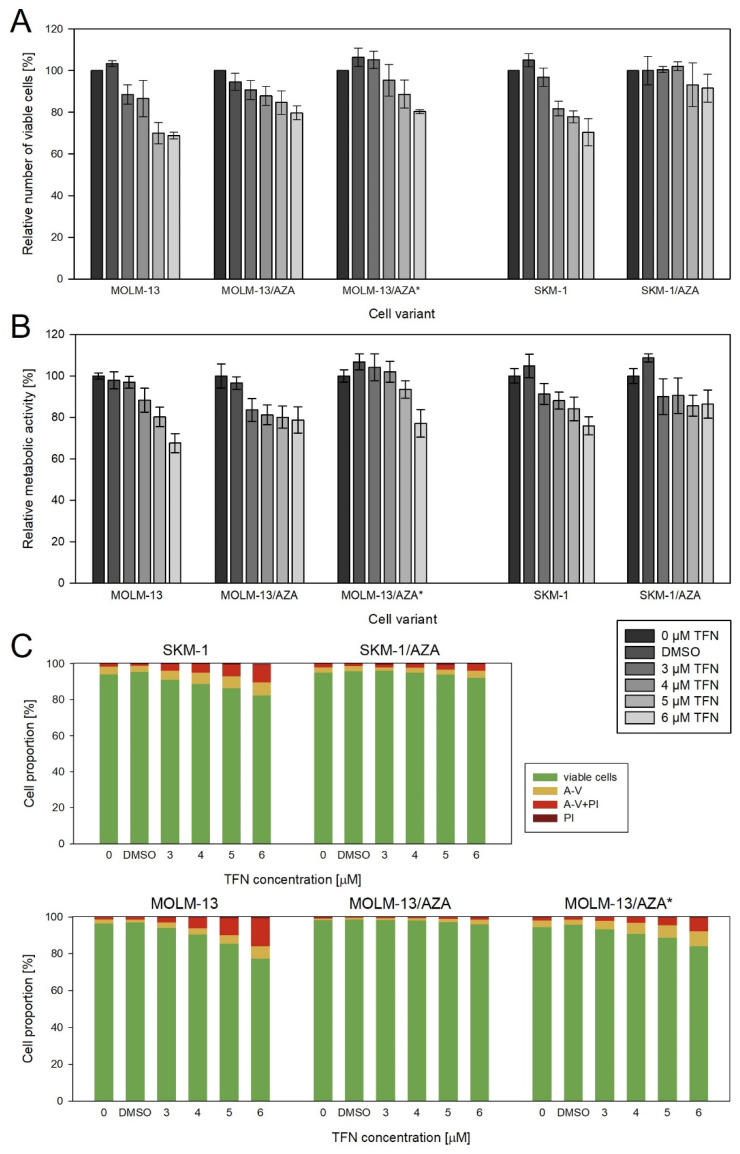
Impact of TFN on AZA-sensitive and AZA-resistant cell variants after 72 h of cultivation. TFN was added to the culture medium every 24 h at four concentrations (3, 4, 5, and 6 μM). Bars for DMSO represent the cells treated with the highest concentration of DMSO present during TFN treatment. (**A**) Relative number of viable cells after TFN treatment compared to untreated controls as measured by CASY TT. (**B**) Relative metabolic activity of cells after TFN treatment as measured by MTS assay. (**C**) Proportions of viable (unstained), apoptotic (stained with Annexin-V or both Annexin-V and PI), and necrotic (stained by PI) cells.

**Figure 7 cancers-15-03063-f007:**
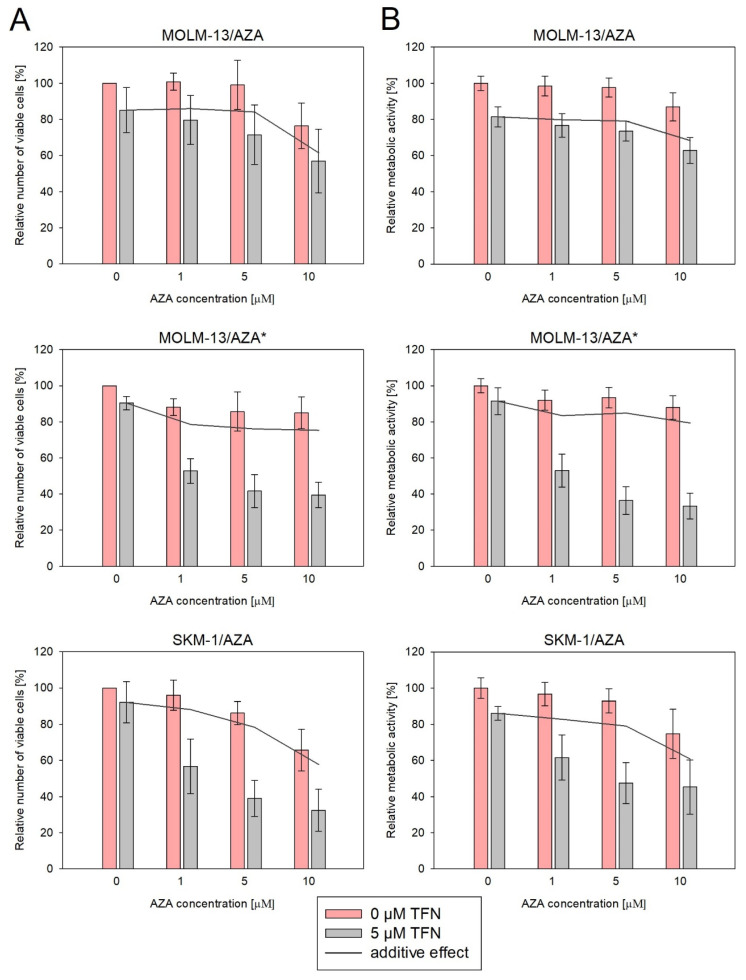
Impact of TFN and AZA cotreatment on AZA-resistant cells after 72 h of cultivation. AZA was added at 1, 5, and 10 μM concentrations to the cells in the presence (gray bars) or absence (pink bars) of 5 μM TFN. Both drugs were added to the culture medium every 24 h. The gray line represents the calculated additive effect of the two drugs. If there is synergy between the drugs, the pink bars should be below the grey line. In the case of a non-synergistic effect, pink bars could appear either above the grey line in the case of an antagonistic effect, or close to the grey line in the case of an interdependent effect of the drugs. (**A**) Relative number of viable cells after AZA and TFN combined treatment compared to cells treated by only one of the drugs as measured by CASY TT. (**B**) Relative metabolic activity of cells after AZA and TFN combined treatment compared to cells treated by only one of the drugs as measured by MTS assay.

**Figure 8 cancers-15-03063-f008:**
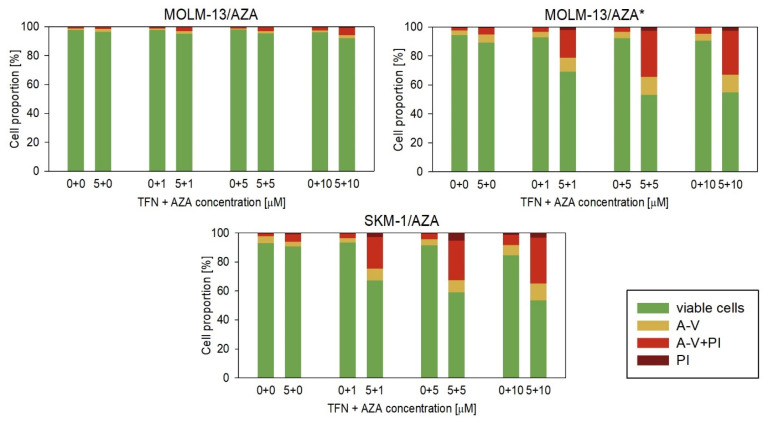
Proportions of viable (unstained), apoptotic (stained with Annexin-V or both Annexin-V and PI), and necrotic (stained by PI) cells after TFN and AZA combined treatment of AZA-resistant cells after 72 h cultivation compared to cells treated by only one of the drugs. AZA was added at concentrations of 1, 5, and 10 μM to the cell culture medium with or without 5 μM TFN. Both drugs were added to the culture medium every 24 h.

**Figure 9 cancers-15-03063-f009:**
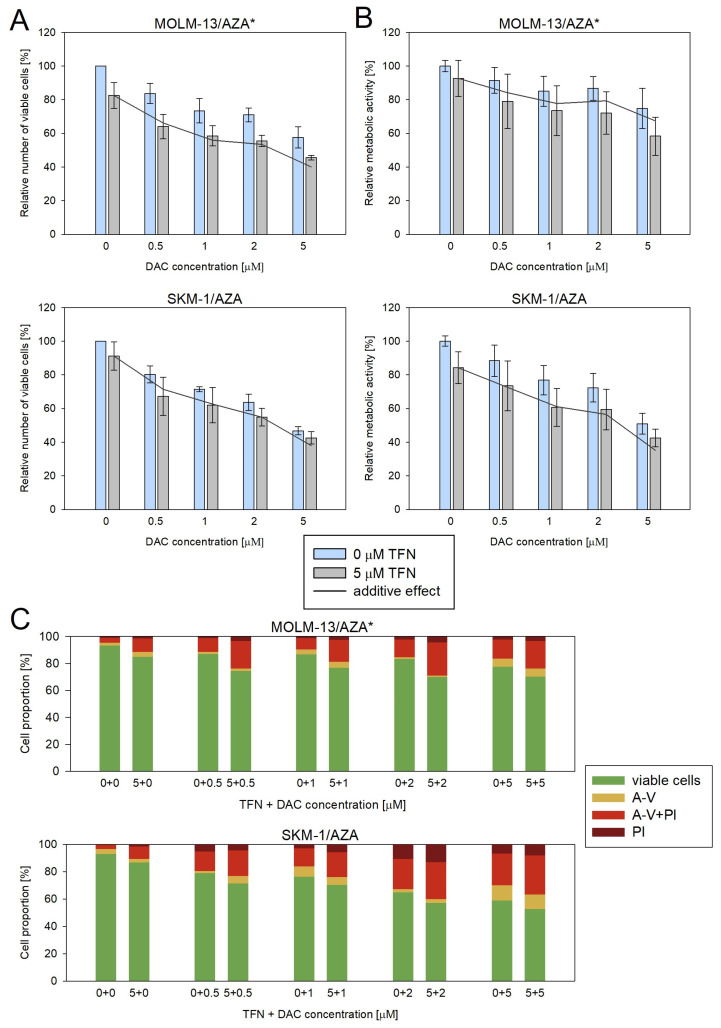
Impact of TFN and DAC combined treatment on AZA-resistant cells after 72 h of cultivation. DAC was added at 0.5, 1, 2, and 5 μM concentrations to the cells in the presence (gray bars) or absence (blue bars) of 5 μM TFN. Both drugs were added to the culture medium every 24 h. The gray line represents the calculated additive effect of the two drugs. If there is synergy between the drugs, the blue bars should be below the grey line. In the case of a non-synergistic effect, blue bars could appear either above the grey line in the case of an antagonistic effect, or close to the grey line in the case of an interdependent effect of the drugs. (**A**) Relative number of viable cells after DAC and TFN combined treatment compared to cells treated by only one of the drugs as measured by CASY TT. (**B**) Relative metabolic activity of cells after DAC and TFN combined treatment compared to cells treated by only one of the drugs as measured by MTS assay. (**C**) Proportion of viable (unstained), apoptotic (stained with Annexin-V or both Annexin-V and PI), and necrotic (stained by PI) cells after DAC and TFN combined treatment compared to cells treated by only one of the drugs.

**Table 1 cancers-15-03063-t001:** PCR primers for respective genes.

Gene	Primer Sequences (5′-3′)	T_A_ (°C)	PCR Product Size (bp)
*ACTB*	CTGGGACGACATGGAGAAAA AAGGAAGGCTGGAAGAGTGC	54.4	564
*CDA*	GCAACATAGAAAATGCCTGCT TAGCAATTGCCCTGAAATCC	56	102
*DCK*	GTCTCAGAAAAATGGTGGGAATG ACAGGTTTCTCTGCATCTTTGAG	56	150
*UCK1*	CGTGTGTGAGAAGATCATGG TGGTCAAAATTGTACTGTCCTTT	56	150
*UCK2*	GACATCAGCGAGAGAGGCAG TCTTGCGTGAAGGGGTGTAG	56	244
*NT5C3A*	ACAACATAGCATCCCCGTGT TTCCTCAAGGCACCATCATGT	58	198
*RRM1*	TGGAATTGGGGTACAAGGTC GAGAGCCCTCATAGGTTTCG	56	176
*RRM2*	TTTACACTGTGATTTTGCTTGC TGTTCTATCCGAACAGCATTG	56	102
*RRM2B*	ATAAACAGGCACAGGCTTCC GAACCTGCACCTCCTGACTAA	58	186
*SAMHD1*	GTTGCCAGTGCTAAACCCAAA TTTCTGTCTGCACACCACTGA	56	293
*SLC29A1*	GTGTCCTTGGTCACTGCTGA GATGCAGGAAGGAGTTGAGG	56	166
*SLC29A2*	ATCCTGAGCACCAACCACAC GTTGAGGAGGGTGAAGAGCA	56	102
*SLC28A1*	AGGTTCTGCCCATCATTGTC CAAGTAGGGCCGGATCAGTA	gradient **	197
*SLC28A3*	GACTCACATCCATGGCTCCT TTCCAGGGAAAGTGGAGTTG	gradient **	183
*SLC28A3* [15]	GAAACATGTTTGACTACCCACAG GTGGAGTTGAAGGCATTCTCTAAAACGT	gradient **	481
*DHODH*	CTGAACACCTGATGCCGACT CCGTAACCTGTGTTCCACCA	58	371
*DCK **	CAGGATCTGGCTTAGCGG CATTTGGCTGCCTGTAGTCT	63	914
*UCK1 **	AGATGGCTTCGGCGGGA AGTCCCTGAACACACATGCC	65	890
*UCK2 **	AACCATGGCCGGGGACAG GATGAGCAGTGCCTCCTGAC	65	858

*—primers used for sequence analysis; **—temperatures between 52 and 65 °C were tested.

**Table 2 cancers-15-03063-t002:** Summary of the characteristics of our cell variants after induction of resistance with AZA and the effect of the HMAs on the cell variants.

Cell Variant	MOLM-13/AZA		MOLM-13/AZA*		SKM-1/AZA	
Sensitivity to DAC (AraC and GEM)	sensitive		considerably decreased sensitivity		slightly decreased sensitivity	
Protein levels compared to parental cell lines						
UCK1	wild (homozygote) →		wild (homozygote) ↓		mutation/wild (heterozygote) ↓	
UCK2	mutation (homozygote) ↓		wild (homozygote) ↓		wild (homozygote) →	
DCK	wild (homozygote) →		wild (homozygote) ↓		wild (homozygote) →	
Protein levels of γ-H2AX and abundance of methylated cytosine in DNA in the cell variants after the cultivation with HMAs						
Cultivation in presence of:	AZA	DAC	AZA	DAC	AZA	DAC
5-mC	→	↓	→	→	→	↓
γ-H2AX	↑↑	↑↑↑↑↑	→	→/↑	↑↑	↑↑
Synergic effect of the drugs on the cell variant						
TFN + HMA	✕	N/A	✓	✕	✓	✕

↓ = downregulation, ↑ = upregulation, → = no difference, ✓ = present, ✕ = absent.

## Data Availability

Additional data and the resistant variants of MOLM-13 and SKM-1 cells are available from the authors upon reasonable request.

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
