# Peer review of "Resistance of Leukemia Cells to 5-Azacytidine: Different Responses to the Same Induction Protocol"

_cancers, 2023, doi:10.3390/cancers15113063_

Round 1
Reviewer 1 Report
The authors showed that the same induction in different cell lines lead to different molecular responses. In this alone the results are not surprising given that the initial genetic background is different in the cell lines. Nevertheless, the authors assessed different mechanisms to understand the different responses. Thus, differences in global DNA methylation, protein levels of DNA methyltransferases and phosphorylation of histone H2AX were observed in response to 5‐Azacytidine (AZA) and 5‐aza‐2ʹ‐deoxycytidine (DAC) treatment. Further, the authors evaluated RNA expression and protein expression of several genes involves in the metabolic pathways of AZA and DAC, and therefore the study is relatively comprehensive.
However, no direct mechanisms involved in resistance to AZA were identified.
Nevertheless, some comments are warranted, together with associated changes in the manuscript.
The authors developed resistance to AZA over a 6‐month period with repeated passaging in medium containing AZA in stepwise increasing concentrations beginning at 0.1 nM. However they fail to indicate the maximum concentrations of resistance attained in all cell lines.
The IC50 was computed by nonlinear regression according to Equation 1 in the manuscript. However, the dose response curves for the MTT assay are not provided in annexes. These curves are important to verify if the IC50 values were correctly computed, since viability curves can be notoriously unreliable, and show plateaus. In this case the calculation of IC50 values is not feasible.
The data is well presented but I recommend that all references to cell lines in figures and tables be standardized (e.g. S/A, M/A and M/A*, however see Figure 1, XX axis: Cell variant: change to Cell line; Figure 1, names used: MOLM-13; MOLM-13/AZA; etc, etc. ).
The UCK gene was sequenced and a homozygous point mutation was found in M/A cells and a heterozygous point mutation was found in S/A cells. The authors speculate that this alteration might change enzyme activity. I suggest using in silico tools to assess whether this change is indeed potentially deleterious or not to strengthen their opinion (SIFT, FATHMM, MutationAssessor, PolyPhen-2, CONDEL, MutationTaster, MutPred, Align GVGD, or PROVEAN), otherwise this is totally speculative and should not be given much weight, and the discussion and conclusion on this alteration should be reduced. If benign, the data should be shown in annexes.
In parallel to influx transporters, efflux transporters play a role in drug resistance. AZA has been shown to induce the expression of certain ABC membrane transporters in some studies. ABC transporters are known to play a role in drug efflux and cellular detoxification processes. The induction of these transporters can lead to increased efflux of drugs or other substances from cells, potentially reducing their effectiveness.
Studies have reported that treatment with 5-azacytidine can upregulate the expression of specific ABC transporters, such as ABCB1 (also known as P-glycoprotein or multidrug resistance protein 1) and ABCG2 (also known as breast cancer resistance protein). For example, in certain cancer cell lines, treatment with AZA has been found to increase the expression and activity of ABCB1, which can confer resistance to chemotherapeutic drugs. The authors have not looked specifically to these efflux genes, but can discuss this and evaluate them inf future studies.
Finally according to Figure 1, the IC50 for gemcitabine (GEM) in all cell lines is very low compared to the other drugs. Why is this? The authors scarcely discuss this data. Does this suggest that GEM can be used as an alternative therapeutic option in myeloid malignancies? A short discussion on Gen would enrich the manuscript, taking into account benefits and adverse reactions of all drugs in clinical settings.
Reviewer 2 Report
The authors introduced the different hypomethylating agents using in myelodysplastic neoplasms (MDS) and acute myeloid leukemia (AML) diseases. The outline of the manuscript is straightforward, and each part is well-organized. This work provides new ideas for MDS and AML treatments. Below are some questions/issues about the work.
Major issues:
Page 6, authors only measured the expression level of DNMT1, but there still have other DNA methyltransferase, like DNMT3. Authors should also measure other DNA methyltransferase level by western blot. Did authors measure the level of DNA damage after the HMAs treatment by MS?
Figure 2A and 2B, authors should use the same concentrate of AZA or DAC between DNMT1 WB assay and 5-mC measurement assay.
Why did the authors only use two reagents for the treatment (AZA and DAC) instead of four (GEM and AraC)?
Figure 3A, the authors' determination of gene expression levels using RT-PCR alone is not sufficient. Author should use RT-qPCR and WB to detect the gene expression.
Figure 4B, the authors found 2 mutation in M/A and S/A cell variant but did not discuss the effect of the mutations on these proteins’ activity.
Page 11 and 13, authors should discuss the experimental results of AZA/DAC and TFN combined treatment and give a reasonable hypothesis, not just describe the results.
Minor issues:
Page 1 line 29, authors should mention the abbreviation of uridine‐cytidine kinases 1 and 2 in abstract.
Figure 3D, the error bar of M/A cell variant is to big, authors should repeat this experiment.
Reviewer 3 Report
Reviewer’s Comments
Different AML cell lines were exposed in vitro continuously to 5-azacytidine (5AZA) in order to select clones that exhibit drug resistance. Nucleoside analogues require phosphorylation by kinases in order to be an active inhibitor. The authors identified variants of AML cells that exhibited signs of 5AZA drug resistance and showed signs of deficiency in UCK1 kinase due to a point mutation in this enzyme. The variants exhibited different responses to treatment with 5AZA and the related analogue, 5‐aza‐2ʹ‐deoxycytidine (DAC). In response to AZA and DAC treatment in these cell variants they exhibited various different responses to sensitivity to the antineoplastic activity these cytosine nucleoside analogues, their inhibition of DNA methylation, and inhibition of DNA repair enzyme, H2AX. Inhibition of de novo synthesis of pyrimidine nucleotides by dihydroorotate dehydrogenase by teriflunomide (TFN) in combination with 5AZA exhibited a synergistic antineoplastic action on the 5AZA resistant variants. The manuscript illustrates the complex pharmacological interaction of 5AZA and DAC on AML cells.
Revised Abstract: These AZA‐resistant variants differ in their responses to other deoxycytidine cytosine nucleoside analogs, including 5‐aza‐2ʹ‐deoxycytidine (DAC), (revise sentence)
The abbreviations for the different variants should be defined at the beginning of the results section.
1. Remove from Figure 3B) Summary of the de novo (green) and salvage (red and blue) pathways of CTP and dCTP synthesis] from Fig 3 and make separate figure for these pathways.
2. Page 15 line 373: This sentence needs clarification: “DAC causes a decrease in the viability of these cells at concentrations as low as 1μM; however, even 40 μM DAC does not cause a 50% decrease in viability, and the IC50 value could not be reliably determined.”
3. The data of Figures 1-3. indicate that DAC is a more potent antineoplastic agent than 5AZA.
A comment should be made in the Discussion about relative drug efficacy of these two cytosine nucleoside analogues. The analogues with the highest potency should be given priority for future investigations in cancer therapy.
4. The authors should describe the differences in subtrate specificity of UCK1 and UCK2.
5. The authors should discuss the potential of TFN in combination with 5AZA or DAC for the treatment of AML.
